# FedSumUp: Secure Federated Learning Without Client-Side Training for Resource-Constrained Edge Devices

## Abstract

Horizontal Federated Learning (HFL) enables multiple clients with private data to collaboratively train a global model without sharing their local data. As a research branch of HFL, Federated Data Condensation with Distribution Matching (FDCDM) introduces a novel collaborative paradigm where clients upload small synthetic datasets instead of gradients and parameters. FDCDM faces two key challenges: privacy leakage risk, where synthetic data may leak the privacy of real data; and high computational cost on the client side, which limits the deployment capability of FDCDM on resource-constrained devices. To address these challenges, we propose FedSumUp, an improved FDCDM method. The core designs of FedSumUp include: generating initial data templates based on a Variational Autoencoder (VAE); and migrating the entire synthetic data optimization process to the server side, requiring clients only to upload distilled synthetic data and the mean of raw data features without exposing the original data itself. Experimental results on multiple real-world datasets demonstrate that FedSumUp achieves notable advantages in the following aspects: drastically reducing the visual similarity between synthetic and real data, and effectively resisting membership inference attacks; significantly lowering client-side computational overhead, making it deployable on edge devices. FedSumUp is the first work to systematically analyze privacy risks in FDCDM from the perspective of data similarity, providing a new direction for building efficient and privacy-preserving federated learning frameworks.

## 1 Introduction

Horizontal Federated Learning (HFL) enables collaborative model training on private data, with significant applications in sensitive domains(Yang et al., 2020). A promising communication-efficient paradigm within HFL is Federated Data Condensation with Distribution Matching (FDCDM), where clients synthesize and upload a small dataset instead of model updates(Xiong et al., 2023; Holland et al., 2024). However, our work identifies two critical, unaddressed challenges that prevent the practical deployment of existing FDCDM methods.

First, despite the goal of preserving privacy, prior works fail to systematically analyze the visual and semantic similarity between the synthetic data they generate and the users' real data. Our research demonstrates that this oversight leads to significant sample-level privacy leakage, making clients vulnerable to Membership Inference Attacks (MIA)(Carlini et al., 2022). Second, the process of generating representative synthetic data requires complex optimization on the client's device, imposing a high computational burden that limits feasibility on resource-constrained hardware.

To overcome these limitations, we propose FedSumUp, a novel FDCDM framework designed for security and efficiency. The core workflow of our method is illustrated in Figure 1. FedSumUp introduces a client-training-free paradigm by leveraging a Variational Autoencoder (VAE) to generate privacy-preserving initial data templates and uniquely migrating the entire expensive data optimization process to the server.

**Our Contributions** We are the first to systematically analyze and demonstrate the visual privacy risks in FDCDM, revealing a critical trade-off between privacy and utility in existing methods; we propose FedSumUp, the first client-training-free FDCDM framework that drastically reduces client-side computational overhead by over 90% compared to methods like FedSD2C, making it practical for edge devices; and we show through extensive experiments that FedSumUp achieves a superior balance of utility, privacy, and efficiency, outperforming baselines in both privacy protection and robustness in challenging non-IID settings.

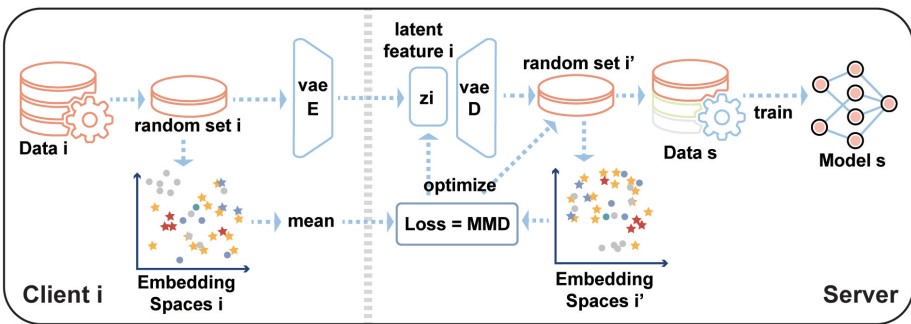

Figure 1: Our proposed FedSumUp method can securely generate a synthetic dataset and conduct most of the computations on the server side. It is used to achieve privacy-preserving federated learning training. Moreover, due to the advantage of "no need for client-side training", it enhances clients' willingness to participate in federated learning.

## 2    RELATED WORK

**Dataset Distillation in Federated learning**    Dataset Distillation (DD) was initially presented in the work of(Wang et al., 2018). Its main objective is to extract the knowledge from datasets and encapsulate it into synthetic data, all the while ensuring that the model trained on this synthetic data maintains a comparable performance to that trained on the original dataset. In the early stages, dataset distillation methods were framed as a bi-level optimization problem(Yu et al., 2023). In this setup, the outer loop focuses on optimizing the synthetic data. This optimization can be achieved through various approaches, such as gradient matching(Zhao et al., 2020; Cazenavette et al., 2022; Yu et al., 2024), distribution matching(Zhao & Bilen, 2023; Wang et al., 2022), and performance matching(Wang et al., 2018). Meanwhile, the inner loop is responsible for incrementally training a model using the synthetic data. Given the limitations of computational resources, single-level optimization methods grounded in kernel ridge regression(Nguyen et al., 2020; 2021) were put forward. These methods aim to break down the bi-level optimization process, thus reducing the overall training cost. On relatively simple datasets like CIFAR10, these methods have shown performance levels comparable to their bi-level counterparts. They have also found applications in Federated Learning (FL), where they are used to tackle problems such as communication bottlenecks(Xiong et al., 2023; Holland et al., 2024), data heterogeneity(Pi et al., 2023; Liu et al., 2022; Zhang et al., 2023), and one-shot FL scenarios(Song et al., 2023; Zhou et al., 2020). Nonetheless, these existing methods demand a substantial amount of computational resources. This makes them unfeasible for use on edge devices in FL, which typically have limited computing power.

**Federated Dataset Condensation with Distribution Matching**    Dataset Condensation with Distribution Matching (DCDM)(Zhao & Bilen, 2023) is a research branch of DD. DCDM optimizes the synthetic data by minimizing the Maximum Mean Discrepancy (MMD) between the synthetic data and the original data. Research(Dong et al., 2022) theoretically demonstrated DCDM's advantages in terms of privacy preservation. Later, FedDM(Xiong et al., 2023) extended the DCDM method from the centralized machine learning setting to HFL, while CollabDM(Holland et al., 2024) and FedSD2C(Zhang et al., 2024) are variants of FedDM designed for One-Shot Federated Learning. And VFDC(Gao et al., 2024) uses this algorithm in Vertical Federated Learning (VFL).However, the study(Zhang et al., 2022) raised concerns regarding the claims made in(Dong et al., 2022), pointing out flaws in the supposed privacy advantages of DCDM. In fact, upon close inspection of the experiments in(Dong et al., 2022), one can observe that the authors used random noise as the initialization template for synthetic data when evaluating similarity between real and synthetic datasets. However, in the utility evaluation, they switched to using real data as the initialization template. This inconsistent setup misleads readers into believing that DCDM simultaneously achieves superior performance in both privacy and utility. Our research demonstrates that this is not the case. Specifically, initializing synthetic data with real images inherently leaks visual privacy information, whereas using random noise for initialization results in significant utility loss.

## 3    PRELIMINARIES

### 3.1    MEMBERSHIP AND VISUAL PRIVACY

Membership Inference Attack(MIA) aims to acquire information about the member samples used for training. MIA can be categorized into three types: those based on Logits scores, those based on labels, and white-box attacks (Li & Zhang, 2025).

In this paper, we introduce an evaluation of FedSumUp's privacy protection level using MIA based on shadow models and inference models (Shokri et al., 2017), which involves two steps: First, the attacker trains shadow models using their own shadow samples (samples similar in type to the real samples) and then trains an inference model based on the outputs of these shadow samples on the shadow model and their membership status. The trained inference model is then used to infer the membership status of target samples held by the attacker.

Additionally, in addition to the MIA mentioned above, our measure of privacy protection also takes into account the visual privacy of synthetic samples. In the study (Dong et al., 2022), "visual privacy" is used to refer to the assessment of similarity between synthetic and real data. In that work, researchers employed L2 distance and LPIPS (Learned Perceptual Image Patch Similarity) to measure this similarity. In addition to quantitative metrics, visual privacy can be subjectively assessed by human observation. Visual privacy and MIA are interconnected; our experiments show that the leakage of visual privacy from synthetic samples can directly aid attackers in performing MIA.

### 3.2 THREAT MODEL

Drawing from the analysis of the aforementioned attack techniques, we define our threat model as follows:

**Adversarial Objective** The primary goal of the adversary is to determine whether a specific data sample was included in the training set. More precisely, given a target sample $x$, the attacker attempts to ascertain whether $x \in T$, where $T$ denotes the original training dataset.

**Adversarial Knowledge** We consider a semi-honest server as the adversary. While the server does not have direct access to the clients' private training data $T$, it possesses full knowledge of the global model parameters and can monitor all communications between the server and client devices. This includes exchanged model updates in frameworks like FedAvg or FedProx, as well as any synthetic data generated during procedures such as those in FedDM.

**Adversarial Capability** The attacker is assumed to have substantial computational resources and access to auxiliary data that closely resembles the distribution of the original dataset $T$. This enables the adversary to construct shadow models and perform visual comparisons or similarity analyses on generated outputs.

### 3.3 VISUAL PRIVACY LEAKAGE IN THE "REAL-DATA MODE"

In the process of applying the DCDM method from machine learning to federated learning, many studies(Xiong et al., 2023; Holland et al., 2024) continue to adopt the pattern design from DCDM, namely the real-data mode and random mode — where real data or random noise is used as the initial template for generating synthetic data. Each client synthesizes $ipc$ pieces of synthetic data for each class of data.

However, our experimental results indicate that the real-data mode can lead to visual privacy leakage. The explanation for each hyperparameter in Table 1 is provided in **Appendix A.5**.

We conducted a visual privacy analysis of the synthetic images generated by DCDM. Table 1 shows the visual results of synthetic data generated for the same class by one client using real data as initial synthetic data template.

Table 1: Visual privacy analysis of DCDM on the CIFAR10 dataset, showing synthetic results for the "dog" class under different settings.

Below Table 1, we provide explanations for each parameter used in the experiments. Based on our experiments, we have the following observations:

1. According to experiments 1-9, it is observed that when the quantity of synthetic images equals that of real images, regardless of the number of epoch iterations performed, the synthetic images completely match the actual data templates of the real images. In scenarios where ipc is equal to or close to $|T|$ (e.g. ipc= $|T|$ = 10), which is common in Non-IID settings within federated learning, the synthetic data uploaded by clients approximates their real data as long as ipc is approximately equal to $|T|$. This approximation violates the principle of preserving privacy in federated learning. In the process of applying the DCDM method to HFL, it is understandable to continue using the IPC design. In DCDM, ipc was designed as a hyperparameter because ipc is much smaller than $|T|$. However, when transferring this design from machine learning to federated learning, it is easy to overlook that the ratio between ipc and $|T|$ has shifted, leading to potential privacy breaches. Our work identifies and verifies this subtle loophole.

2. Based on experiments 11, 14, and 17, we observe that under the condition of $R_s = 0.1$, different values of ipc lead to different visual privacy outcomes. Specifically, keeping $R_s$ constant, generating 1 image at a time results in better visual privacy compared to generating 10 or 100 images at a time.

3. The findings from experiments 12 and 21 reveal that, even when the L2 distance between synthetic images and their corresponding template images is significantly large, the template images remain the closest to the synthetic images compared to other real images. This observation suggests that synthetic images preserve certain characteristics of the original template images. Under edge computing conditions, where the computational power of client devices is limited and restricts the number of synthesis iteration epochs, if a server acts as a semi-honest attacker with access to an initial template sample used for generating synthetic data, it could perform MIA without the need to train shadow models. In this scenario, the server could rely solely on the L2 distance and visual inspection to execute such attacks. These results also highlight that using real data as templates for synthetic data generation is inadequate for ensuring privacy protection.

### 3.4 Utility Loss in the "Random Mode"

Using random noise images as initial templates, DCDM in federated learning leads to low training accuracy and non-convergence issues. Our experiments have demonstrated this point, as shown in Table 2 and Figure 2, which illustrate performance across various dimensions when using random noise versus real data as the initial template for the generation of synthetic data within FedDM(Xiong et al., 2023). The basic experimental parameters are also annotated.

It can be observed that employing random noise as the initial template achieves slightly better privacy protection measured by MIA compared to using real data. However, its performance across other dimensions is inferior. As discussed previously, utilizing real data as the initial template poses visual privacy concerns. Naturally, this prompts us to seek a novel method for generating the initial template for synthetic data.

Table 2: Performance and Configuration of Federated Learning Experiment

(a) Experimental Performance

| Metric | Random Template | Real Template |
|---|---|---|
| ACC | 0.3833 | **0.4495** |
| MIA ACC | **0.4747** | 0.5055 |
| Client Time (s) | 3000 | **1300** |

(b) Experimental Configuration

| FL Algorithm | Dataset | Number of Clients |
|---|---|---|
| FedDM | CIFAR10 | 10 |
| Model | Batch Size | Dirichlet $\alpha$ |
| Convnet | 256 | 0.30 |
| Rounds | Learning Rate | IPC |
| Early Stopping | 0.01 | 5 |
| Synth LR | Synth Epochs | Server Epochs |
| 1 | 200 | 50 |

### 4 Methodology

To address the challenges of privacy leakage and high computational costs in FDCDM, we propose FedSumUp, a novel framework that eliminates the need for client-side training. As illustrated in Figure 1, our approach migrates the entire data synthesis and optimization process to the server, requiring clients to only upload lightweight, privacy-preserving data summaries.

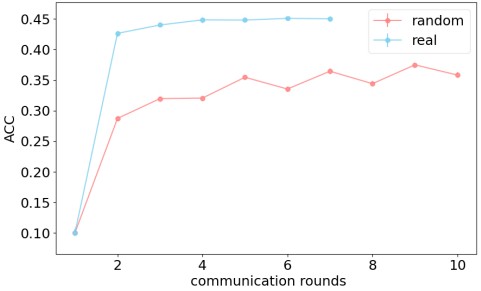

Figure 2: Test accuracy in different communication rounds

## 4.1 OUR INNOVATIONS

FedSumUp is not a simple integration of existing methods but a systematic restructuring aimed at addressing the unresolved privacy-utility-efficiency trilemma in Federated Data Condensation with Distribution Matching (FDCDM). Building upon the research foundations of CollabDM(Holland et al., 2024) and FedSD2C(Zhang et al., 2024), we propose three fundamental innovations:

**Theoretical Innovation** We are the first to systematically reveal the risk of visual privacy leakage in FDCDM, demonstrating that synthetic data generated through real-data templates exhibits a high degree of visual similarity to the original data.

**Framework Innovation** We propose a client-training-free paradigm by migrating the entire synthetic data optimization process to the server side. Clients only need to upload lightweight summary information (VAE latent codes and mean feature vectors), reducing client-side computational load by 90% compared to FedSD2C.

**Algorithmic Innovation** We design a two-phase server-side optimization process, Optimizing VAE latent codes through Maximum Mean Discrepancy (MMD) loss to align the feature distributions of synthetic and real data; Performing refined optimization on pixel-level synthetic data to enhance model utility.

## 4.2 THE FEDSUMUP FRAMEWORK

Our framework consists of two main components: a lightweight client-side process for generating data summaries and a server-side process for data condensation and global model training.

**Client-Side Summary Generation** Instead of performing costly local training, each client executes a simple, one-pass procedure. Given its local data, a client uses a pre-trained Variational Autoencoder (VAE) to encode a small batch of images into compact latent codes. Simultaneously, it calculates the mean of the feature embeddings for these images using the current global model. Finally, the client transmits only these latent codes and the mean feature vectors to the server. This process avoids sharing raw data or gradients, significantly reducing both privacy risks and computational load.

---

**Algorithm 1** FedSumUp Client Algorithm (Per-Round)

---

**Require:**
1: TrainSet: Class-partitioned training data, TrainSet$[c]$ for class $c$;
2: Model: Client-side local ML model;
3: VAE: Variational autoencoder for data encoding;
4: $x$: Random set from TrainSet$[c]$;
5: $y$: Labels of $x$;
6: ipc: Number of samples in $x$, ipc $= |x|$;
7: Round: Current round number;
8: GlobalModel: Server-sent global model.
9: **if** Round $== 1$ **then**
10:     Model $\leftarrow$ RandomInitialize(Model)
11: **else**
12:     Model $\leftarrow$ GlobalModel
13: **end if**
                                                             ▷ Data Processing:
14: **for all** $c \in$ Classes **do**
15:     $x \leftarrow$ RandomSelection(TrainSet$[c]$, ipc)
16:     $y \leftarrow$ GetLabels($x$)
17:     cTensor $\leftarrow$ encode(VAE, $x$)
18:     $F_d \leftarrow$ embed(Model, $x$)
19:     MeanOfFeature $\leftarrow \frac{1}{|F_d|} \sum F_d$
20:     CoreTensor $\leftarrow$ CoreTensor $\cup \{$cTensor$\}$
21:     MeanOfFeatures $\leftarrow$ MeanOfFeatures $\cup \{$MeanOfFeature$\}$
22:     Labels $\leftarrow$ Labels $\cup \{y\}$
23: **end for**
                                                             ▷ Data Transmission:
24: Send $\{$CoreTensor, Labels, MeanOfFeatures$\}$ to server

---

**Server-Side Data Condensation and Global Training** The server receives the compact summaries from all participating clients. It then performs a two-phase optimization process to generate

a small, high-fidelity synthetic dataset. In the first phase, it optimizes the latent codes to align the feature distribution of the decoded data with the mean feature vectors provided by the clients. In the second phase, it directly optimizes the pixel values of the decoded images for further refinement. Once this high-quality synthetic dataset is generated and aggregated, it is used to train the global model for the next round.

---

**Algorithm 2** FedSumUp Server Algorithm (Per-Round)

---

**Require:**
    **Model**: Server-side machine learning model;
    VAE: A variational autoencoder used for data encoding;
    $\eta_c$: Learning rate for updating $z_i$;
    $\mathcal{S}_i$: A set of condensed data for $client_i$;
    $S$: A set that accumulates condensed data from all clients;
    $Y$: Labels for $S$;
    $M_F$: Mean of features extracted from the original client data;
    $F_i$: Features of the generated samples using the global model;
    $E_1$: Epochs to optimize $z_i$;
    $E_2$: Epochs optimizing $S_i$ after decoding.

                                                            ▷ Merge and Data Condensation Process:

1: **for all** $client_i \in$ clients **do**
2:    $M_{Fi} \leftarrow$ MeanOfFeatures from $client_i$
3:    $z_i \leftarrow$ CoreTensor from $client_i$
4:    $y_i \leftarrow$ Labels from $client_i$

                                               ▷ (Phase 1): Optimize latent code $z_i$

5:    **for** epoch $= 0$ to $E_1$ **do**
6:       $S_i \leftarrow$ decode(VAE, $z_i$)
7:       $F_i \leftarrow$ embed(Model, $S_i$)
8:       $L = \left\| M_{Fi} - \frac{1}{|F_i|} \sum F_i \right\|^2$
9:       $z_i \leftarrow z_i - \eta_c \nabla_{z_i} L$
10:   **end for**

                                         ▷ (Phase 2): Decode and optimize sample $S_i$

11:   $S_i \leftarrow$ decode(VAE, $z_i$)
12:   **for** epoch $= 0$ to $E_2$ **do**
13:      $F_i \leftarrow$ embed(Model, $S_i$)
14:      $L = \left\| M_{Fi} - \frac{1}{|F_i|} \sum F_i \right\|^2$
15:      $S_i \leftarrow S_i - \eta_c \nabla_{S_i} L$
16:   **end for**
17:   $S$ **append** $S_i$
18:   $Y$ **append** $y_i$
19: **end for**
20: Train Model using $\{S, Y\}$
21: Send updated Model to all clients

---

The detailed step-by-step breakdown of both client and server algorithms are provided in **Appendix A.1.**

## 5 EXPERIMENT

The parameters used in this experimental section are as follows:

**Federated Learning (FL) Algorithms:** Six FL algorithms are compared, categorized into three types: model parameter uploading methods FedAvg (McMahan et al., 2017), FedProx (Li et al., 2020); federated data condensation methods FedDM (Xiong et al., 2023), FedSD2C (Zhang et al., 2024), CollabDM (Holland et al., 2024); and methods for resource-constrained devices CollabDM (Holland et al., 2024).

**Datasets:** Three commonly adopted datasets are utilized: CIFAR10, MNIST, and FashionMNIST.

**Experimental Setup:** A total of 10 clients participate in the federated learning process by default. The experiments are conducted using the ConvNet architecture, with each client employing a batch size of 256 during training. Data heterogeneity among clients is controlled by a Dirichlet distribution with $\alpha = 0.5$, indicating moderate non-IID (non-independent and identically distributed) characteristics. The server-side learning rate for training the global model is set to 0.01.

**Training Termination:** The number of communication rounds is determined by an early stopping mechanism. For multi-round algorithms, training is terminated if the global model's accuracy improves by less than 1 percent compared to the previous round, ensuring efficient resource utilization for constrained clients.

**Evaluation Metrics:** Accuracy (ACC) and Membership Inference Attack Accuracy (MIA ACC) are reported with four decimal places, while total client training time is presented with two significant figures.

**Special Configurations:** FedDM and CollabDM use real data as synthesis templates. In the proposed method, the number of images per class (ipc) is set to 300 by default, and the number of optimization steps for synthesizing data is set to 200.

More detailed experimental configurations can be found in the default settings of the submitted source code repository. We analyzed the sensitivity of key hyperparameters, including IPC, E1, and E2, to validate our model's stability. The full analysis is provided in **Appendix A.2**.

## 5.1 BASELINES

Our proposed method, FedSumUp, demonstrates superior performance across multiple datasets, including MNIST, FashionMNIST, and CIFAR10, compared to other baseline approaches.

Table 3: Baselines

| Dataset | Method | ACC | MIA ACC | Client Time(s) | Visual Privacy Leakage |
|---|---|---|---|---|---|
| MNIST | FedAvg | 0.9847 | 0.6508 | 2000 | - |
| | FedProx | 0.9794 | 0.6473 | 2700 | - |
| | FedDM | 0.9864 | 0.6416 | 240 | True |
| | FedSD2C | **0.9886** | 0.5355 | 3100 | False |
| | CollabDM | 0.9439 | 0.6337 | 280 | True |
| | **FedSumUp(Ours)** | 0.9883 | **0.5076** | **190** | False |
| FashionMNIST | FedAvg | **0.8954** | 0.6349 | 2000 | - |
| | FedProx | 0.8891 | 0.6388 | 2100 | - |
| | FedDM | 0.8676 | 0.6337 | **240** | True |
| | FedSD2C | 0.872 | 0.4764 | 3300 | False |
| | CollabDM | 0.8420 | 0.5607 | 300 | True |
| | **FedSumUp(Ours)** | 0.8541 | **0.4543** | **240** | False |
| CIFAR10 | FedAvg | **0.6311** | 0.58 | 3400 | - |
| | FedProx | 0.6026 | 0.568 | 3000 | - |
| | FedDM | 0.5911 | 0.5488 | 330 | True |
| | FedSD2C | 0.5927 | 0.5223 | 2500 | False |
| | CollabDM | 0.3876 | **0.4421** | **280** | True |
| | **FedSumUp(Ours)** | 0.5621 | 0.5138 | **280** | False |

The results, summarized in Table 3, highlight FedSumUp's advantages. On all datasets, our method achieves a balance of utility, privacy, and efficiency. Notably, on MNIST, FedSumUp reduces client-side time to just 190s (a 10x improvement over FedAvg) and lowers MIA ACC to a near-random 0.5076, demonstrating strong privacy preservation with competitive model accuracy. While FedSD2C achieves comparable privacy and utility, our approach is over 15x more efficient on the client-side.

## 5.2 SYNTHETIC DATA VISUALIZATION

In this section, we present a comprehensive visualization of the synthetic data generated by FedSumUp, aiming to provide both global and local insights into how the synthetic samples relate to the original client data. This helps us understand not only the overall distributional alignment between real and synthetic data but also the fine-grained structural differences that are critical for privacy preservation.

Figure 3 presents a UMAP projection of the original and synthetic data from the FashionMNIST dataset. In sub-figure (a), which corresponds to the initial stage where only the latent space is optimized, we observe a noticeable discrepancy between the distributions of the original and synthetic data. This indicates that the synthetic samples have not yet closely aligned with the structure of the true data.

However, in sub-figure (b), after further optimizing both the latent representations and the decoded pixel values, the synthetic data becomes more tightly clustered around the original data points, showing improved alignment in terms of global distribution and class-wise clustering patterns. This demonstrates the effectiveness of our multi-stage optimization strategy in enhancing the realism of synthetic data while preserving semantic consistency.

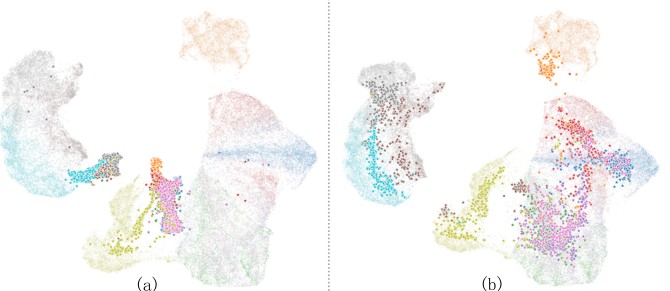

(a)    (b)

Figure 3: UMAP visualization for FashionMNIST. The semi-transparent circles represent the distribution of original data, and the opaque pentagrams represent the distribution of synthetic data. (a) shows the data distribution after optimizing only the latent space, and (b) shows the distribution after further optimizing both the latent space and the decoded pixel values.

While the UMAP plot offers a macro-level view of the data distribution, it does not reveal how individual synthetic samples compare to their original counterparts at the image level. To address this, we conduct a fine-grained visual comparison in Table 4, focusing on several representative classes from the FashionMNIST dataset.

In this experiment, $S_0$ represents synthetic data, $T_0$ denotes the original data corresponding to the synthetic data, and $T_v$ signifies the intermediate process data obtained after encoding the original data, optimizing it, and then decoding. Further optimization of $S_0$ is performed using $T_v$. $T_{L2}$ refers to the data with the smallest L2 distance from $S_0$ among all similar data.

The purpose of presenting this table is to demonstrate that when the server receives synthetic data generated by the FedSumUp algorithm, it cannot accurately determine whether its own samples are included in the synthetic data using the methods mentioned earlier. This effectively resists membership inference attacks (MIA) conducted by the server through visual privacy.

Table 4: synthetic and original data visualization for FashionMNIST

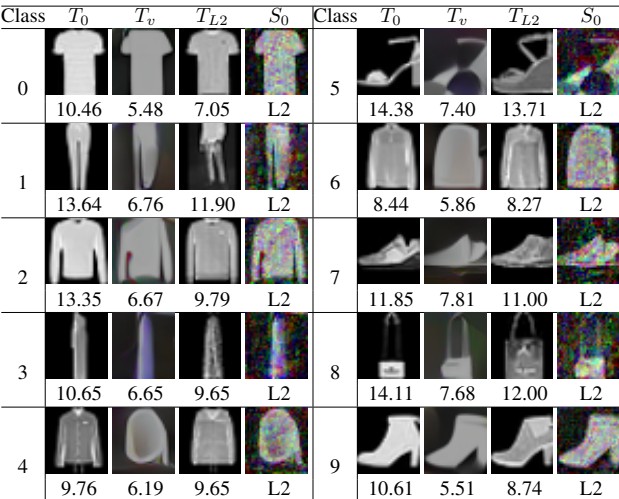

Additionally, this table intuitively explains why FedSumUp can resist MIA based on training shadow models. The principle of MIA is to judge whether a sample was used for training based on abnormal outputs from the trained model. In other words, for a data sample $D_i$, its personalized features (i.e., features not present in other data of the same category as $D_i$) assist attackers in conducting MIA. From the four images in class 8, we can observe that the personalized patterns in the center of the handbag (patterns not possessed by all bags) are eliminated after the FedSumUp synthetic data process. This indicates that the FedSumUp process helps remove personalized features of the data while retaining common features beneficial for image recognition among similar data. Consequently, it achieves significant enhancement in privacy protection against MIA with minimal loss of utility.

This theory also explains why FedSumUp exhibits a greater decline in utility performance compared to FedAvg on CIFAR10 relative to MNIST and FashionMNIST. Since CIFAR10 is more complex than MNIST and FashionMNIST, during the VAE encoding, optimization, and decoding processes in

FedSumUp, not only are the personalized features that leak privacy removed, but also some common features of the data are eliminated.

## 5.3 NON-IID STUDY

Table 5: Performance comparison of different methods under varying Non-IID settings on the CIFAR-10 dataset.

| alpha | Method | ACC | MIA ACC | Client Time(s) | Visual Privacy Leakage |
|---|---|---|---|---|---|
| 0.01 | FedProx | 0.0994 | **0.4697** | 470 | - |
| | FedDM | 0.4809 | 0.525 | **65** | True |
| | **FedSumUp(Ours)** | **0.5401** | 0.499 | 130 | False |
| 0.1 | FedProx | 0.2794 | **0.4825** | 1500 | - |
| | FedDM | **0.5331** | 0.5383 | 200 | True |
| | **FedSumUp(Ours)** | 0.5289 | 0.5071 | **130** | False |
| 0.3 | FedProx | **0.6083** | 0.5732 | 2900 | - |
| | FedDM | 0.5959 | 0.5586 | 320 | True |
| | **FedSumUp(Ours)** | 0.581 | **0.5093** | **260** | False |
| 0.5 | FedProx | **0.6026** | 0.568 | 3800 | - |
| | FedDM | 0.5911 | 0.5488 | 330 | True |
| | **FedSumUp(Ours)** | 0.5621 | **0.5138** | **280** | False |
| 1 | FedProx | **0.6571** | 0.5884 | 3300 | - |
| | FedDM | 0.6339 | 0.5714 | **250** | True |
| | **FedSumUp(Ours)** | 0.5883 | **0.5213** | 370 | False |

The performance under varying degrees of non-IID data is shown in Table 5. Our method, Fed-SumUp, demonstrates superior robustness in challenging scenarios. For example, in the extreme non-IID setting ($\alpha = 0.01$), FedSumUp's accuracy of 0.5401 significantly surpasses both FedProx (0.0994) and FedDM (0.4809). Furthermore, it consistently maintains low MIA ACC across all settings, preserving client privacy. In terms of efficiency, FedSumUp strikes an excellent balance, avoiding the prohibitive computational costs incurred by methods like FedProx in heterogeneous environments (e.g., 3800s at $\alpha = 0.5$).

## 5.4 ADDITIONAL EXPERIMENTS

**Ablation Study** To validate our key design choices, we conducted a comprehensive ablation study. The results demonstrate that migrating the optimization process to the server-side is the most critical factor for reducing client-side computation, achieving a 90% time reduction with minimal impact on model accuracy. The detailed step-by-step analysis is presented in **Appendix A.3.**

**Scalability** We also verified our method's scalability with up to 100 clients. FedSumUp consistently maintained superior efficiency and privacy protection as the number of clients grew, outperforming baselines like FedAvg and FedDM in balancing these metrics. Detailed scalability results are available in **Appendix A.4.**

## 6 CONCLUSION AND FUTURE WORKS

We introduced FedSumUp, a novel client-zero-training framework for FDCDM that resolves the critical privacy-utility dilemma in prior works. By migrating optimization to the server, FedSumUp protects user privacy and drastically reduces client computation while maintaining high utility and non-IID robustness. Future work will explore more advanced privacy attacks and the design of personalized VAEs, as well as promote the application of FedSumUp in text processing tasks and large model domains. We also advocate for the systematic evaluation of data similarity in all future synthetic-data-based FL research.Our future directions include exploring more attack methods to evaluate privacy preservation in federated data condensation, training personalized VAEs with improved privacy-utility trade-off objectives, and encouraging systematic evaluation of synthetic-real data similarity in synthetic-data-based federated learning; additionally, we highlight two guiding principles for federated data condensation: shifting computation to the server under resource constraints and enabling user-defined privacy with provider-defined utility to balance privacy and performance.

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

# A APPENDIX

## A.1 DETAILED ALGORITHMS AND METHODOLOGY

**Model Initialization or Update** In the first round, the client initializes its local model randomly. In subsequent rounds, it receives and adopts the updated global model from the server.

**Class-wise Data Processing** For each class $c$, a subset $x$ of size ipc is randomly sampled from TrainSet[c] with corresponding labels $y$. The samples $x$ are encoded via a Variational Autoencoder (VAE) into compact latent representations stored in cTensor. The original samples $x$ are passed through the local model, and feature embeddings are extracted using the embed function; the average feature vector per class is computed and stored in MeanOfFeatures. True labels $y$ are collected directly.

**Aggregation** Encoded tensors, averaged features, and labels are aggregated into three lists: CoreTensor, MeanOfFeatures, and Labels, respectively.

**Data Transmission** After processing all classes, the client sends the aggregated data {CoreTensor, Labels, MeanOfFeatures} to the central server. No raw data, gradients, or model parameters are transmitted, enhancing privacy and reducing bandwidth consumption.

**Data Condensation Process (per client)** For each client $client_i$, the server retrieves its feature mean $M_F$, core tensor $z_i$, and labels $y_i$. In Phase 1, the latent code $z_i$ is optimized over $E_1$ epochs by minimizing the discrepancy between the original feature mean $M_F$ and the average feature representation of decoded samples:

$$L = \left\| M_{Fi} - \frac{1}{|F_i|} \sum F_i \right\|^2$$

where $F_i$ is obtained by embedding the decoded samples $S_i = \text{decode}(\text{VAE}, z_i)$ through the global model. The latent code is updated via gradient descent:

$$z_i \leftarrow z_i - \eta_c \nabla_{z_i} L$$

In Phase 2, after decoding $z_i$ into $S_i$, the synthesized samples are refined over $E_2$ epochs using the same loss function, optimizing the pixel space directly:

$$S_i \leftarrow S_i - \eta_c \nabla_{S_i} L$$

**Aggregation** After refining each client's latent code and generating synthetic samples $S_i$, the server aggregates all samples into a unified dataset $S$ and their corresponding labels into $Y$.

**Global Model Training** The global model is trained on the aggregated condensed dataset $\{S, Y\}$, distilling knowledge from all clients without accessing private data. After training, the updated global model is broadcast back to clients for the next communication round.

## A.2 HYPERPARAMETER ANALYSIS

In this section, we analyze the impact of three key hyperparameters—IPC (Images Per Class), E1 (VAE encoding optimization rounds), and E2 (synthetic data optimization rounds)—on model performance using the CIFAR10 dataset. The goal is to clarify how these parameters affect accuracy, privacy risk (measured by membership inference attack accuracy, MIA ACC), and computational cost on the client side.

### A.2.1 IPC (IMAGES PER CLASS)

IPC refers to the number of synthetic images generated per class per client. As shown in Figure 4, increasing IPC generally leads to improved model accuracy (ACC). This is expected, as more synthetic samples provide richer information for model training, enhancing generalization ability. On the other hand, the MIA ACC remains relatively stable across different IPC values, indicating that changes in IPC do not significantly affect the privacy leakage risk. However, a higher IPC value increases the client-side computation time due to the larger volume of data that needs to be processed and optimized.

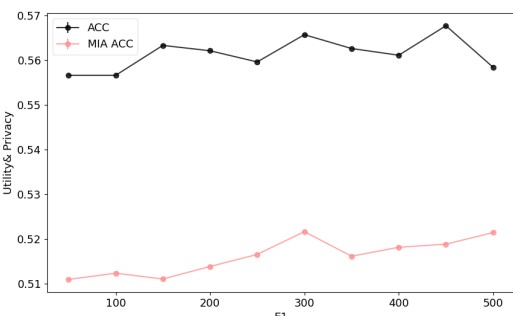

Figure 4: Impact of IPC on ACC, MIA ACC, and client-side processing time.

### A.2.2 E1 (VAE ENCODING OPTIMIZATION ROUNDS)

E1 denotes the number of optimization iterations applied during the encoding phase of the variational autoencoder for synthetic data generation. As illustrated in Figure 5, experimental results indicate that both accuracy and membership inference attack accuracy fluctuate with changes in E1 values. Certain E1 values can achieve optimal accuracy (e.g., E1=450), which suggests that moderate encoding optimization can improve the quality of latent representations, thus benefiting downstream learning tasks. Similarly, membership inference attack accuracy also varies with E1, indicating that the encoding optimization process impacts privacy protection.

Figure 5: Impact of E1 on ACC and MIA ACC.

### A.2.3 E2 (SYNTHETIC DATA OPTIMIZATION ROUNDS)

E2 indicates the number of optimization iterations directly applied to the synthetic data after decoding from the variational autoencoder. As depicted in Figure 6, our experiments show that both accuracy and membership inference attack accuracy fluctuate with E2. Accuracy reaches a local maximum when E2 is 150, whereas privacy leakage risk peaks at E2=400.

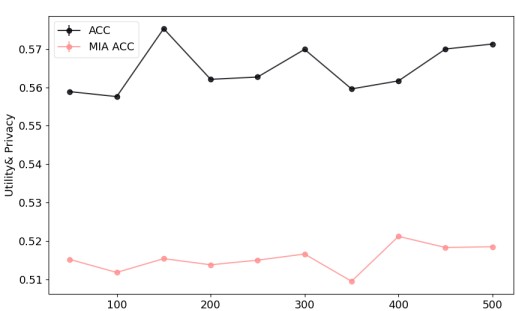

Figure 6: Impact of E2 on ACC and MIA ACC.

A.3   ABLATION STUDY

To validate the effectiveness of each improvement in **FedSumUp**, we start from the original method **FedSD2C** and progressively introduce different modules. We compare their performance on the CIFAR-10 dataset. Through this series of ablation experiments, we analyze the impact of each key component on model accuracy (ACC), privacy risk measured by membership inference attack accuracy (MIA ACC), and client-side computational cost (Client Time).

1. **Original FedSD2C Method**
   As a baseline, FedSD2C performs local training on clients and uses soft labels with KL divergence loss for knowledge distillation. Synthetic data generation relies on core sample selection, Fourier perturbation mechanism, and latent space optimization of the VAE encoder on the client side.

2. **Remove Client Training + Use Hard Labels + Replace KL Loss with Cross-Entropy (CancelTrain)**
   We first removed the local model training process on the client side, replaced soft labels with hard labels, and substituted the KL divergence loss used during server-side training with cross-entropy loss. This change reduced client-side computation but led to a drop in overall accuracy (from 0.5927 to 0.4934), indicating that local training still has some value.

3. **Random Selection Instead of Core Set Selection (RandomSet)**
   Replacing the importance-based core sample selection strategy with random sampling resulted in improved performance (ACC = 0.5157). This suggests that the core sample selection has little effect on synthetic data quality.

4. **Remove Fourier Perturbation (NoFourier)**
   Removing the Fourier noise perturbation mechanism slightly increased model accuracy (ACC = 0.5181), while decreasing MIA accuracy. This indicates that the perturbation affects model learning to some extent but helps reduce privacy leakage risk.

5. **Move Synthetic Data Optimization to Server Side (ServerSideOpt)**
   Transferring the originally client-side synthetic data optimization to the server side significantly reduced client-side runtime (down to only 85s), without sacrificing model accuracy (ACC = 0.5187).

6. **Add Direct Optimization in Image Space (ImageOpt)**
   Building upon latent space optimization in the VAE, we added direct pixel-level optimization on the decoded image space, which further improved synthetic data quality while maintaining stable model performance.

7. **Multi-Round Synthetic Data Optimization (Final Version: FedSumUp)**
   Finally, based on all the above improvements, we extended single-round synthetic data optimization into a multi-round iterative framework, forming the complete **FedSumUp** method. It achieves a high accuracy of **0.5621**, maintains low privacy risk (MIA ACC = 0.5138), and keeps client-side time at a relatively low level (280s).

This ablation study demonstrates that FedSumUp is formed through systematic improvements over FedSD2C. These enhancements include removing client-side training, adopting hard labels and cross-entropy loss, using random sample selection, eliminating Fourier perturbation, centralizing optimization tasks on the server, introducing image-space optimization, and applying multi-round synthesis strategies. As a result, FedSumUp emerges as a new federated learning approach that balances client efficiency, strong model accuracy, and robust privacy protection.

Table 6: Performance comparison of different variants of FedSD2C on CIFAR-10, including accuracy (ACC), membership inference attack accuracy (MIA ACC), and client-side execution time in seconds.

| Progress | ACC | MIA ACC | Client Time(s) |
|---|---|---|---|
| FedSD2C | 0.5927 | 0.5223 | 3100 |
| CancelTrain | 0.4934 | 0.5103 | 2700 |
| RandomSet | 0.5157 | 0.5176 | 2700 |
| NoFourier | 0.5181 | 0.5076 | 2700 |
| ServerSideOpt | 0.5187 | 0.5088 | 85 |
| ImageOpt | 0.5137 | 0.5107 | 90 |
| **MultiRounds(FedSumUp)** | 0.5621 | 0.5138 | 280 |

## A.4 SCALABILITY STUDY

In this experiment, we evaluated the performance of three methods (FedAvg, FedDM, and Fed-SumUp) under different numbers of clients, focusing on three key metrics: model accuracy (ACC), privacy protection level (Privacy Score), and average training time per client (Client Time). The experiments were conducted with varying numbers of clients: 1, 10, 30, 50, and 100, to study the scalability of the algorithms under different system loads.

Table 7: Complete scalability study results.

| No. of Clients | Method | ACC | MIA ACC | Client Time(s) | Visual Privacy Leakage |
|---|---|---|---|---|---|
| 1 | FedAvg | **0.7934** | 0.6839 | 1000 | - |
| | FedDM | 0.4773 | 0.5216 | **40** | Ture |
| | **FedSumUp(Ours)** | 0.5286 | **0.5114** | 150 | False |
| 10 | FedAvg | 0.6334 | 0.5794 | 3400 | - |
| | FedDM | 0.5911 | 0.5488 | 320 | Ture |
| | **FedSumUp(Ours)** | 0.5621 | **0.5138** | 280 | False |
| 30 | FedAvg | 0.525 | 0.545 | 1500 | - |
| | FedDM | **0.6854** | 0.5932 | 1000 | Ture |
| | **FedSumUp(Ours)** | 0.5929 | **0.5232** | **440** | False |
| 50 | FedAvg | 0.5984 | 0.5703 | 2500 | - |
| | FedDM | **0.7114** | 0.6202 | 1800 | Ture |
| | **FedSumUp(Ours)** | 0.5814 | **0.5173** | **560** | False |
| 100 | FedAvg | 0.6091 | 0.5761 | 4300 | - |
| | FedDM | **0.731** | 0.6395 | 2300 | Ture |
| | **FedSumUp(Ours)** | 0.5771 | **0.5241** | **730** | False |

From the results, it can be observed that as the number of clients increases, FedAvg exhibits significant fluctuations in accuracy and a notable increase in client training time, indicating potential performance bottlenecks in large-scale deployments. FedDM performs well in terms of accuracy, achieving the highest ACC value with a larger number of clients. However, its Privacy Score is relatively high, implying weaker privacy protection. Moreover, the client training time of FedDM is significantly higher than that of FedSumUp. Additionally, as previously discussed, FedDM suffers from visual privacy leakage in realistic settings, which poses a critical drawback in terms of privacy preservation. In contrast, our proposed method, FedSumUp, significantly outperforms the other two methods in terms of Client Time while maintaining a low Privacy Score, demonstrating its advantage in balancing efficiency and privacy protection. Although the accuracy of FedSumUp is slightly lower than that of FedDM, its overall comprehensive performance makes it more suitable for deployment in real-world large-scale distributed environments.

## A.5 EXPLANATION OF HYPERPARAMETERS FOR TABLE 1

Table 8: Explanation of hyperparameters used in the visual privacy analysis (Table 1).

| Parameter | Explanation |
|---|---|
| $i$ | Experiment ID |
| $Epoches$ | How many steps the optimizer iterates over the dataset to synthesize images |
| $R_s$ | The proportion of synthesized images relative to real images |
| $T$ | The set of real images used for synthesizing images, with all images in $T$ belonging to the same category |
| $|T|$ | The size of the set T |
| $ipc$ | Synthesized Image per Class (IPC), defined as $ipc = |T| \times R_s$, indicating the number of synthesized images generated for each class. |
| $S_0$ | The synthesized image with index 0, referring to the first synthesized image in the sequence |
| $T_0$ | The real image used as a template to synthesize $S_0$ |
| $T_{L2}$ | The image within T that is closest to $S_0$ in terms of L2 distance |
| $|L2|$ | Measures the L2 distance between $T_{L2}$ and $S_0$ |

## A.6 VAE MODEL GENERALITY AND PRE-TRAINING

To clarify the nature of the Variational Autoencoder (VAE) used in the FedSumUp framework, we address a potential question regarding its implementation and dataset dependency.

**Question:** Is the VAE model used in FedSumUp specifically trained or fine-tuned on the downstream task datasets (e.g., CIFAR10, MNIST)? Does using this VAE introduce additional training overhead on the client side or require knowledge of the client's data distribution beforehand?

**Answer:** No. The VAE employed in our framework is a **general-purpose, pre-trained model** that is entirely independent of the specific datasets used in our federated learning experiments. We use a single, fixed VAE that was trained beforehand on a large-scale, public image dataset, and its weights remain frozen throughout all experiments.

This design is a deliberate and critical aspect of FedSumUp for two primary reasons:

- **Alignment with Zero Client-Side Training:** Our core contribution is to eliminate the training burden on client devices. Requiring clients to train or fine-tune a VAE would contradict this principle. By providing a ready-to-use, pre-trained VAE, we ensure the client-side process remains lightweight, involving only a simple one-pass encoding.

- **Enhanced Privacy and Generality:** Training a VAE on specific client data could inadvertently cause it to memorize features of that private data, creating a potential privacy risk. Using a universal, pre-trained VAE avoids this issue. It acts as a common, public feature transformation tool that does not depend on any single client's data distribution, thereby strengthening the privacy-preserving nature of our framework.

In summary, the VAE functions as a universal data encoder and decoder. Its parameters are not part of the federated learning updates and are not adapted to any specific dataset, which reinforces the efficiency, security, and scalability of our proposed method.

## A.7 Statement on the Use of Large Language Models (LLMs)

In compliance with the ICLR 2026 policy, we report the use of Large Language Models (LLMs) as assistive tools in the preparation of this paper. The specific roles of the LLM are detailed below:

- **Manuscript Polishing and Refinement:** The LLM was employed to improve the overall quality of the writing. This included correcting grammatical errors, rephrasing sentences for better clarity and flow, and ensuring consistent terminology throughout the document. It served as an advanced grammar and style checker.

- **Algorithm Code Implementation:** The LLM was used to assist in the practical implementation of our proposed algorithms. Its functions included generating boilerplate code for data handling, translating high-level pseudocode into Python snippets, and offering suggestions for debugging specific, isolated code segments.

We emphasize that all text and code generated by the LLM were critically reviewed, verified, and extensively revised by the authors to ensure scientific accuracy, correctness, and originality. The authors retain full responsibility for all content presented in this paper.

