# OpenReview forum: "FedSumUp:Secure Federated Learning Without Client-Side Training for Resource-Constrained Edge Devices"
_ICLR.cc/2026/Conference — Submitted to ICLR 2026_

### Official Review · Reviewer_FmzS · 2025-10-29

**Soundness:** 3
**Presentation:** 2
**Contribution:** 3
**Rating:** 4
**Confidence:** 4

**Summary:**

This paper introduces FedSumUp, a federated data condensation framework designed specifically to address privacy leakage and client-side computational overhead issues in horizontal federated learning. Instead of requiring clients to synthesize and optimize data locally, FedSumUp shifts all expensive data optimization to the server and has clients only upload VAE-encoded latent codes and mean data features, thereby sharply reducing computational load and limiting privacy exposure. The paper systematically critiques current FDCDM paradigms, exposing visual privacy leakage in real-data initialization and utility degradation under random-noise initialization, and presents extensive experiments to demonstrate improved privacy, efficiency, and performance under various non-IID settings compared to several strong baselines.

**Strengths:**

1) The paper is the first to rigorously expose and analyze visual privacy leakage in FDCDM schemes, especially under real-data initialization. This is well-illustrated with Table 1 and the corresponding discussion on Page 4, and visually connected to MIA vulnerabilities.
2)  By offloading all synthetic data optimization to the server, the proposed method massively reduces resource requirements on clients (as substantiated by Tables 3, 5, 6, and 7). Table 3 highlights that client runtime is reduced by over 10–15x compared to other methods.
3) The paper proposes a clever privacy-preserving mechanism. It uses a general-purpose, pre-trained VAE as a privacy filter, where clients upload highly abstract "latent codes" rather than raw images. The server first optimizes these codes in the latent space before decoding them. This process tends to filter out personalized features while retaining the common class features beneficial for model training , thereby mechanistically helping to resist Membership Inference Attacks (MIA).

**Weaknesses:**

1. While the paper exposes practical visual privacy leakages in prior FDCDM methods, its claimed privacy enhancements in FedSumUp are predominantly empirical (via MIA ACC, Table 3 and Table 5). There is no formal privacy analysis or theoretical bound (e.g., differential privacy guarantees, or information-theoretic leakage quantification).
2. While Appendix A.6 claims that the VAE is universal and not fine-tuned per client or task, the actual privacy and generalization performance of this VAE is not deeply interrogated. What happens if the VAE is insufficiently expressive for specific domains or tasks? Could the VAE itself encode subtle privacy leakages if, for example, the upstream training dataset for the VAE includes client-resembling data? No empirical test of VAE generality or security robustness is attempted.
3. The protocol seems to assume honest-but-curious clients, but if clients upload malicious codes or manipulated means, what prevents poisoning or information leakage back to the server or other clients? There is no discussion of potential mechanisms.
4. While MNIST, FashionMNIST, and CIFAR10 are standard, they are relatively small and may not sufficiently represent real-world high-heterogeneity, high-dimensional, or non-vision FL tasks. It remains unclear if the claimed privacy/utility gains hold for domains outside canonical image datasets.

**Questions:**

See Weakness.

---

### Official Review · Reviewer_A8Df · 2025-10-30

**Soundness:** 2
**Presentation:** 2
**Contribution:** 2
**Rating:** 4
**Confidence:** 3

**Summary:**

This paper identifies two critical challenges in the existing Federated Data Condensation with Distribution Matching (FDCDM) paradigm: 1) significant privacy risks when using real data as templates for synthetic data generation, and 2) high computational costs on the resource-constrained edge device client side. To address these issues, the paper proposes FedSumUp, in which each client sends (per-class) VAE latent codes and mean feature vectors; the server performs a two-phase optimization (latent → pixel) to synthesize a global, small dataset used to train the global model.

**Strengths:**

1. By offloading all complex optimization to the server, the client-side burden is reduced to a simple one-pass encoding and feature extraction.

2. By using a VAE to generate abstract latent codes, FedSumUp avoids using templates that are either too realistic (leaking privacy) or too noisy (hurting utility).

**Weaknesses:**

1. The server must now perform a two-phase optimization (latent code and pixel-level) for each participating client in every round. This cost could be substantial and scales with the number of clients, yet it is not reported, which makes the "efficiency" claim one-sided.

2. The paper assumes a semi-honest server adversary and evaluates privacy against server-side MIA, yet the server receives per-class latent codes and mean feature vectors every round. The server with the public VAE decoder may decode latents to image-like content. No further analysis is provided on how much information these latents/means reveal.

**Questions:**

1. What is the total computational overhead on the server, and how does this cost scale as the number of clients increases?

2. The entire method relies on a general-purpose VAE pre-trained on a public dataset. How would the method's performance be affected if the clients' private data comes from a highly specialized domain that is significantly "out-of-distribution" for the VAE's pre-training data?

3. You opted for a weaker optimization objective on the server to match the mean of the real and synthetic features, rather than a stronger distributional loss. Was this choice primarily for efficiency?

4. What is the reconstruction fidelity when directly decoding transmitted latents with the provided/public decoder?

5. If the server actively optimizes latents to probe the client distribution, how robust is FedSumUp to targeted reconstruction/inversion?

---

### Official Review · Reviewer_LYj3 · 2025-11-01

**Soundness:** 2
**Presentation:** 1
**Contribution:** 2
**Rating:** 2
**Confidence:** 3

**Summary:**

This paper extends Federated Data Condensation with Distribution Matching (FDCDM) by addressing privacy limitations and computational constraints on edge devices. The authors incorporate Variational Autoencoders (VAE) to extract latent representations of client-side data, which are then transmitted to the server for synthetic data generation. This approach requires only serverside model training, thereby reducing the computational burden on clients and mitigating sample-level privacy leakage by transferring latent representations instead of ”initial templates” that could result in visual information leakage.

**Strengths:**

The paper presents a complete framework with improvements in accuracy and computational efficiency compared to baseline methods.

**Weaknesses:**

1. The paper suffers from significant organizational issues that impede comprehension. While the work builds upon Heterogeneous Federated Learning (HFL), this foundational design choice is not clearly articulated. The framework is only briefly mentioned at the beginning of the Introduction, using vague terminology to describe the challenges and background of HFL before transitioning to FDCDM and data security concerns. This fragmented presentation makes it difficult for readers to understand the core methodology and its relationship to existing work.

2. Additionally, the paper’s contributions are presented in an unprofessional manner, lacking sufficient evidence to support claimed security benefits. The computational advantage is also inadequately substantiated, with only a vague claim of ”reducing client side computational overhead by over 90% compared to methods like FedSD2C,” which lacks rigor and clarity.

3. The rationale for incorporating VAE to prevent sample leakage is inadequately explained. While the paper dedicates considerable space to discussing how the initial template used for synthetic data generation poses a risk of privacy leakage through potential attacks, it fails to provide a clear explanation of why and how VAE addresses this vulnerability. The connection between the VAE-based latent representation and enhanced privacy protection remains unclear.

4. Despite claiming ”theoretical innovation,” the paper provides no theoretical analysis or formal results. The absence of theoretical foundations significantly weakens the paper’s contributions and makes it difficult to assess the principled nature of the proposed approach.

5. The experimental evaluation is inadequate to support the paper’s claims. The experiments are limited to simple datasets and moderately non-IID settings, which is insufficient to demonstrate the method’s effectiveness. The performance evaluation does not adequately explore more challenging tasks or varied heterogeneous settings, making it less convinced regarding the improvement.

**Questions:**

See weakness.

---

### Official Review · Reviewer_VTkQ · 2025-11-03

**Soundness:** 3
**Presentation:** 3
**Contribution:** 3
**Rating:** 4
**Confidence:** 3

**Summary:**

This paper tackles two core weaknesses in Federated Data Condensation with Distribution Matching (FDCDM), a branch of horizontal federated learning:

1. Synthetic datasets can still resemble real client data.
2. Existing FDCDM algorithms need heavy local optimization.

Authors present the first systematic privacy-risk analysis in FDCDM from a data-similarity viewpoint.They propose FedSumUp, where a pre-trained VAE is used to create initial synthetic templates, all expensive optimizations are shifted to the server, leaving only data summarization task on clients.

Experimental Results show the following:
Far less visual similarity between synthetic and real data → improved privacy.
High reduction in client computation, making it very suitable for edge devices.

**Strengths:**

+ Systematically reduction of visual similarity between synthetic and real data
+ No direct data, gradients, or parameters ever leave the client.
+ No Client-Side Training
+ VAE based summarization provides a standardized, privacy-safe feature extraction pipeline
+ Centralizing the condensation and MMD-based alignment process ensures consistent optimization quality and reduces heterogeneity issues caused by varying client compute capacities.
+ Balanced Utility and Privacy

**Weaknesses:**

- Performance heavily relies on the representational strength and generalization of the pre-trained VAE. If the VAE does not capture key features relevant to a domain (e.g., medical images), synthetic data quality may degrade.
- Migrating all optimization to the server increases centralized computational load, which can become difficult with large datasets and large number of clients.
- Since the method removes personalized patterns to prevent MIAs, models may lose subtle but useful client-specific features, affecting tasks that rely on personalization
- Since VAE is available to the server, can't the data be recovered through gradient inversion?
- Centralizing all optimization increases the risk of server compromise; a malicious server could still attempt to reverse-engineer latent features.
- It hasn't been tested on real-world images yet. The datasets used are very basic ones and less challenging.
- More exhaustive experiments and real-world setups of FL should be explored as done in the following paper (inspired by Office 31):
"Federated Learning for Commercial Image Sources", WACV 2023.
Dataset link: https://drive.google.com/file/d/1qgpj1TsGT4lnhhOmwR4gqVRigoHnMRnX

**Questions:**

Although raw data is not shared, mean feature embeddings might still reveal distributional hints of private data. How to handle that?

---

### Meta-Review · Area_Chair_YZiE · 2025-12-08

**Summary:**

The paper proposes FedSumUp, a Federated Learning framework that leverages Federated Data Condensation with Distribution Matching (FDCDM) and Variational Autoencoders (VAE) to reduce client-side computational costs and mitigate privacy risks. By offloading synthetic data optimization to the server and transmitting only latent data and mean feature vectors, the method aims to be suitable for resource-constrained edge devices.

While reviewers acknowledged the strengths of the proposed method in reducing client-side computation and improving empirical privacy, they pointed out several critical weaknesses. The primary concerns include the heavy reliance on a pre-trained VAE, which may not generalize to specialized domains, the lack of theoretical privacy guarantees, and the significant increase in server-side computational load. Reviewers also questioned the robustness against model inversion attacks, given that the server has access to the VAE decoder and latent codes. The experimental evaluation was deemed insufficient, limited to simple datasets (MNIST, CIFAR-10) without exploring more challenging real-world scenarios or larger models.

Given the consistent negative comments and the absence of an author response, I recommend rejection.

**Reviewer Concerns:**

Not applicable because the authors did not submit a rebuttal.

**Reviewer Scores:**

All the reviewers will maintain their scores because the author did not submit a rebuttal.

---

### Decision · Program_Chairs · 2026-01-26

Reject